# Enhancing Salt Tolerance in Poplar Seedlings through Arbuscular Mycorrhizal Fungi Symbiosis

**DOI:** 10.3390/plants13020233

**Published:** 2024-01-14

**Authors:** Shuo Han, Yao Cheng, Guanqi Wu, Xiangwei He, Guozhu Zhao

**Affiliations:** 1College of Biological Sciences and Technology, Beijing Forestry University, Beijing 100083, China; holahan99@bjfu.edu.cn (S.H.); chengyao102100@163.com (Y.C.); wgq5248@163.com (G.W.); 2National Engineering Research Center of Tree Breeding and Ecological Restoration, Beijing Forestry University, Beijing 100083, China

**Keywords:** arbuscular mycorrhizal fungi, poplar, salt stress, physiological mechanism, partial least squares path modeling

## Abstract

Poplar (*Populus* spp.) is a valuable tree species with multiple applications in afforestation. However, its growth in saline areas, including coastal regions, is limited. This study aimed to investigate the physiological mechanisms of arbuscular mycorrhizal fungi (AMF) symbiosis with 84K (*P. alba × P. tremula var. glandulosa*) poplar under salt stress. We conducted pot experiments using NaCl solutions of 0 mM (control), 100 mM (moderate stress), and 200 mM (severe stress) and evaluated the colonization of AMF and various physiological parameters of plants, including photosynthesis, biomass, antioxidant enzyme activity, nutrients, and ion concentration. Partial least squares path modeling (PLS-PM) was employed to elucidate how AMF can improve salt tolerance in poplar. The results demonstrated that AMF successfully colonized the roots of plants under salt stress, effectively alleviated water loss by increasing the transpiration rate, and significantly enhanced the biomass of poplar seedlings. Mycorrhiza reduced proline and malondialdehyde accumulation while enhancing the activity of antioxidant enzymes, thus improving plasma membrane stability. Additionally, AMF mitigated Na^+^ accumulation in plants, contributing to the maintenance of a favorable ion balance. These findings highlight the effectiveness of using suitable AMF to improve conditions for economically significant tree species in salt-affected areas, thereby promoting their utilization.

## 1. Introduction

Escalating soil salinity represents a crucial factor impeding plant productivity. Soil salinization is a growing problem worldwide that occurs due to overuse, irrigation, and clearing [1]. High concentrations of sodium ions in saline soils hinder plant water and nutrient absorption [2]. This water scarcity and nutrient deficiency lead to osmotic and ionic stress in plants, resulting in oxidative stress and other secondary stresses [3]. Ultimately, this stress can lead to physiological drought, stunted growth, reduced yield, and even plant death [4]. Consequently, addressing the increasingly severe threat of soil salinization is imperative to the sustainable development of agriculture and forestry.

Modern energy and timber production rely heavily on forests [5]. Poplar, as one of the important tree species undergoing afforestation, exhibits wide distribution across diverse climatic regions, contributing to substantial physiological diversity and notable variability in salt tolerance among trees [6]. Due to its rapid growth and high biomass production, it has become an important species for afforestation and agroforestry projects around the world [7]. The escalating salinization of soil has placed considerable constraints on the cultivation of salt-sensitive poplar species [8]. Salt stress can inhibit poplar growth, induce photosynthesis and system damage, and change plants’ physiological and cellular morphology [9]. Additionally, poplar possesses abundant genomic information and molecular tools, making it a valuable model tree species for investigating plant–microorganism interactions [10]. Based on the response of poplar to soil salinity at the cellular and whole-plant levels, poplar can be used as a model species to elucidate the physiological and molecular mechanism of tree stress resistance [11]. Enhancing the salt tolerance of poplar is important for the development of contemporary forestry.

Arbuscular mycorrhizal fungi (AMF) are an important component of soil ecosystems and are widely distributed. They can form symbiotic relationships with the roots of higher plants, thereby playing a crucial regulatory role in ecosystems. Extensive research has been conducted across various crops to understand how AMF enhances plant salt tolerance. AMF can enhance plant salt tolerance through several mechanisms, including improved nutrient acquisition, enhanced antioxidant enzyme activity, reduced reactive oxygen species levels, and the maintenance of water status. In addition, AMF can induce a molecular mechanism for improving the salt tolerance of the host through gene expression changes (*PIP*, Na/H antiporters, *LSNCED*, *LSLEA*, and *LSP5cs*) [12,13]. A previous study indicated that the salt tolerance of *Gleditsia sinensis* was significantly enhanced through AMF inoculation in a greenhouse setting, attributed to increased antioxidant enzyme activity and the alleviation of ionic toxicity [14]. In another study, *Olea europaea* L. inoculated with AMF successfully maintained high stem water potential levels by upregulating its primary and secondary metabolites, thereby significantly reducing the detrimental effects of water deficit stress [15]. Furthermore, AMF inoculation has been observed to regulate phosphate and water uptake, thus enhancing the drought tolerance of *Populus × canadensis ‘Neva’* under drought stress conditions [16]. It is noteworthy that AMF significantly improved the root morphology and biomass of *Gleditsia sinensis* under salt stress [17]. AMF primarily alleviated the adverse effects of salt stress on AMF-inoculated *Zelkova serrata* seedlings by reducing Na^+^ content, increasing P, K^+^, and Mg^2+^ concentrations, and enhancing leaves’ photosynthetic levels and biomass accumulation [18]. The dual mycorrhizal association with ectomycorrhizal fungi (ECM), arbuscular mycorrhizal fungi, and non-mycorrhizal endophytes (FE) enhances poplar stress resistance [19]. It is important to note that ECM dominates in more mature trees, whereas AMF is present throughout the growth stages of poplar, especially at the seedling stage [20]. Despite AMF holding great potential for enhancing salt tolerance in woody plants, our comprehension of how AMF influences the growth and salt tolerance of poplar remains limited. This study aims to elucidate the physiological mechanisms through which AMF enhances plant salt tolerance, offering a theoretical foundation for afforestation and restoration applications.

Here, we focus on 84K poplar, a male interspecific hybrid resulting from a breeding program led by Professor SinKyu Hyun (Seoul National University, Korea). This hybrid exhibits rapid growth and excellent adaptability to various environments, making it suitable for short-rotation cultivation purposes such as timber and firewood production [21]. This study is based on the hypothesis that AMF promotes the growth and salt tolerance of poplar seedlings. To verify the effect of AMF on salt tolerance in the leaves and roots of poplar, we inoculated 84K poplar with AMF (*Rhizophagus irregularis*) in a pot designed to detect AMF colonization and changes in the physiological indexes of the plant (plant growth quality, nutrient content, antioxidant defense, osmotic balance, and photosynthetic physiology). We analyzed the effects of AMF on poplar’s metabolic and chemical defense systems and reveal the underlying physiological mechanisms. Through the modeling of multivariate relationships, our results provide new information on the complex mechanism of AMF in enhancing the salt tolerance of poplar. This study serves as a theoretical basis for understanding how AMF can be used to enhance chemical defense in poplar and provides valuable insights into this field of research.

## 2. Results

### 2.1. Mycorrhizal Colonization

At all NaCl concentrations, the indigenous AMF in the nonsterilized soil exhibited a colonization rate of approximately 24% on the plant roots (Figure 1b). Seedlings inoculated with AMF demonstrated a considerable abundance of vesicles, and the mycorrhizal colonization rate reached 70% across all NaCl levels. However, the AMF colonization rate showed a significant 10% reduction at a 200 mM NaCl concentration when compared to the rates at 0 and 100 mM. The reduction in the mycorrhizal colonization rate was markedly influenced by the salt concentration and AMF inoculation (*p* < 0.001). Notably, a significant interaction effect between salt concentration and AMF inoculation was observed (Appendix A). While salt stress did lead to a reduction in soil spore density, it did not reach a statistically significant level (Figure 1c).

### 2.2. Soil GRSP Content under Salt Stress

The variations in salt concentration did not yield statistically significant changes in glomalin-related soil protein (GRSP) levels. In soil samples not inoculated with AMF, the contents of easily extractable glomalin-related soil protein (EEG) and total glomalin-related soil protein (TG) were 4.07 and 1.92 mg/g, respectively (Figure 2). In contrast, the contents of EEG and TG in AMF-inoculated soil were significantly increased, with levels of 13.38 and 6.52 mg/g, respectively. Although the increase in salinity demonstrated a slight inhibitory effect on GRSP content, this effect did not attain statistical significance (Appendix A). The analysis of variance showed a significant influence of AMF inoculation on GRSP levels (*p* < 0.001).

### 2.3. Plant Growth Quality

With the escalation of salt stress intensity, a notable decreasing trend occurred in both the shoot and root biomass of poplar under different treatments, along with a reduction in the root/shoot ratio (Appendix A). Inoculation with AMF led to a significant increase in biomass and the root/shoot ratio compared to non−inoculated plants. Salt stress had a pronounced negative effect on plant relative water content (RWC), but AMF inoculation effectively increased RWC levels and alleviated the negative effect of salt stress. Salt stress induced a significant increase in lignin and cellulose concentrations in poplar stems, and AMF inoculation further increased these concentrations compared to non−inoculated plants. The analysis of variance showed significant effects of both salt stress and AMF inoculation on plant biomass and the root−to−shoot ratio and a significant interaction between salt stress and AMF inoculation (Appendix A). Leaf RWC was affected by the interaction between salt stress and AMF inoculation. Stem lignin and cellulose contents were significantly affected by salt stress and AMF inoculation but did not demonstrate a response to the interaction between these factors.

### 2.4. Leaf Photosynthesis and Chlorophyll Fluorescence Parameters

With the increase in salt concentration from 0 mM to 100 mM and 200 mM, the net photosynthetic rate (Pn), intercellular CO_2_ concentration (Ci), and transpiration rate (Tr) exhibited a decline, reaching a remarkably low level under severe stress conditions (Appendix A). Under non−stress conditions, AMF inoculation led to a substantial increase in plant Ci. In the presence of salt stress, AMF inoculation had a slight enhancing effect on both Pn and Ci. Regardless of stress conditions, plants inoculated with AMF exhibited significantly boosted Tr compared to non−inoculated plants. Both salt concentration and AMF inoculation significantly influenced these photosynthetic parameters (*p* < 0.001) (Appendix A).

Without AMF inoculation, the photochemical quenching coefficient (qP), non−photochemical quenching coefficient (qN), maximum photochemical quantum yield of the PSII reaction centers (Fv/Fm), and actual quantum yield of the PSII (Y(II)) of plants gradually decreased with increasing salt stress intensity (Appendix A). Under severe salt stress, plants inoculated with AMF demonstrated a significant increase in qN, Fv/Fm, and Y(II) compared to non−inoculated plants. An analysis of variance revealed significant effects of both salt stress and AMF inoculation on qN, Fv/Fm, and Y(II) (*p* < 0.001), with the interaction between salt stress and AMF inoculation significantly affecting the Fv/Fm of plants (Appendix A).

### 2.5. Antioxidant Defense of Plants

As salt stress intensity increased, the malondialdehyde (MDA) content in plants exhibited a gradual increase, while AMF inoculation consistently led to a considerable reduction in MDA content across varying NaCl concentrations. In comparison to non−inoculated seedlings, leaves and roots displayed an average decrease of 22% and 21%, respectively (Figure 3a). A two−way analysis of variance demonstrated that plant MDA content was significantly affected by salt stress, AMF inoculation, and their interaction (Appendix A). Simultaneously, with the increase in salt concentration, the activities of superoxide dismutase (SOD), peroxidase (POD), and catalase (CAT) in plant leaves and roots showed an increasing trend (Figure 3b–d). Under salt stress conditions, AMF−inoculated plants exhibited significantly elevated SOD activity, resulting in an average increase of 7% in leaves and 8% in roots when compared to non−inoculated seedlings. AMF inoculation also led to notably higher CAT and POD activities in leaves, regardless of salt concentration. However, in roots, these enzyme activities were significantly higher than in non−inoculated plants only when exposed to 200 mM NaCl. Following inoculation with AMF, the CAT activity in leaves increased by 63%, the CAT activity in roots increased by 26%, the POD activity in leaves increased by 13%, and the POD activity in roots increased by 6%. A two−way analysis of variance indicated that plant CAT activity was significantly affected by both salt stress and AMF inoculation, but not by their interaction (Appendix A). POD activity was significantly influenced by both salt stress and AMF inoculation, with the interaction only affecting the enzyme activity in roots. The activity of SOD in plants was significantly influenced by salt stress, AMF inoculation, and their interaction.

### 2.6. Absorption of Elements and Ions in Plants

The concentrations of C and N in both the shoot and root parts of the plants exhibited significant reductions under salt stress conditions (Figure 4a,b). In contrast, samples inoculated with AMF exhibited a notable increase in C and N content compared to non−inoculated samples. The shoot and root C content increased by 11% and 20%, respectively, and the N content increased by 4% and 16%, respectively. Plant C and N were influenced by both salt stress and AMF inoculation and their interaction had a significant impact on plant C and underground N content (Appendix A). The P content of the roots decreased with the increase in salt stress, while the P content of the shoots decreased significantly at a concentration of 200 mM NaCl (Figure 4c). AMF inoculation effectively mitigated the suppression of P content induced by salt stress, resulting in a 6% increase in shoot P content and a substantial 21% increase in root P content. Salt stress significantly affected P content. AMF inoculation had an effect on P content in the whole plant, while the interaction between salt stress and AMF inoculation had no significant effect (Appendix A).

As salt concentration increased, the Na^+^ concentration also increased, with an average elevation of 125% in the aboveground part and 249% in the underground part (Figure 4d). Compared with non−inoculated plants, AMF inoculation led to a significant reduction in Na^+^ concentration within plant tissues under salt stress, resulting in an average decrease of 39% in shoots and an average reduction of 18% in roots. An analysis of variance showed that salt stress, AMF inoculation, and their interaction played a critical role in determining plant Na^+^ concentrations (Appendix A). The K^+^ concentration in shoots significantly decreased under 200 mM salt concentration and increased following AMF inoculation (Figure 4e). The root K^+^ concentration decreased under salt stress and significantly increased due to AMF inoculation, with an average increases of 8% and 37% in shoots and roots, respectively. Salt stress and AMF inoculation had a substantial influence on plant K^+^, with the interaction between the two factors only affecting root K^+^ (Appendix A). Under salt stress, the Ca^2+^ concentration increased. At a 200 mM salt concentration, AMF inoculation increased the shoot and root Ca^2+^ concentrations (Figure 4f), with an overall increase of 13% in shoots. Salt stress, AMF inoculation, and their interaction had a significant effect on plant Ca^2+^ concentration (Appendix A). The Mg^2+^ concentration in plants decreased at 200 mM salt concentration, but this decrease was mitigated by AMF inoculation (Figure 4g). Salt stress affected the underground Mg^2+^ content, and the interaction between AMF inoculation and salt stress had a notable impact on the overall Mg^2+^ concentration in the plants (Appendix A).

### 2.7. Osmotic Balance in Plants

Increasing salt concentration significantly accelerated the relative electrolyte leakage (REL) in plant leaves (Figure 5a). AMF inoculation resulted in a remarkable reduction in REL at 200 mM NaCl compared to the no−inoculation treatment. Both salt stress and AMF inoculation had substantial effects on REL (Appendix A). As the salt concentration increased, the content of proline (Pro) and glycine betaine (GB) in plants gradually increased (Figure 5b,c). Compared to the no−inoculation treatment, AMF inoculation significantly increased the Pro content in leaves and roots by 10% and 22%, respectively, and boosted the GB content in leaves and roots by 21% and 78%, respectively. Salt stress and AMF inoculation had substantial effects on plant Pro and GB content, with a notable interaction effect on Pro content (Appendix A). Salt stress also resulted in an increase in the content of soluble sugars and sucrose in plants (Figure 5d,e). AMF inoculation enhanced soluble sugar content under salt stress. Both salt stress and AMF inoculation had notable effects on plant soluble sugar and sucrose content, and the interaction between salt stress and AMF inoculation significantly affected soluble sugar content (Appendix A).

### 2.8. PCA Analysis of Plants Inoculated with AMF under Salt Stress

To investigate whether plants exhibit different physiological response patterns to salt stress and mycorrhizal inoculation, principal component analysis (PCA) was performed using the experimental dataset. PCA reduces the dimensions of all variables to the most representative dimensions, transforming large datasets into representative scores for each sample. The PCA results revealed that principal component 1 (PC1) and principal component 2 (PC2) accounted for 55.98% and 26.11% of the variance, respectively, collectively explaining 82.09% of the total variance (Figure 6). An increasing trend in salt concentration was observed along PC1, while mycorrhizal inoculation showed distinct separation along PC2.

### 2.9. Partial Least Squares Path Modeling (PLS−PM) of AMF’s Effect on Salt Tolerance of Poplar

To further analyze the role of mycorrhizae in the salt tolerance of poplar trees, a partial least squares path model was constructed for analysis. In the aboveground part of the plants, mycorrhiza enhance the actual photosynthetic electron transport efficiency and transpiration efficiency, thereby improving photosynthetic capacity (Figure 7). They also reduce the Na^+^/K^+^ ratio, thus enhancing salt tolerance and subsequently increasing the aboveground biomass. In the underground part, mycorrhiza regulate osmotic balance through the modulation of proline and glycine betaine, leading to enhanced photosynthetic capacity and salt tolerance adjustment. This, in turn, results in increased root biomass.

## 3. Discussion

Popular is a valuable tree species with economic significance, but salt stress adversely affects its physiology, growth, and productivity. Throughout the course of evolution, poplar trees have developed various mechanisms to adapt to salt stress [22]. Mycorrhizal symbiosis plays a crucial role in establishing a mutualistic relationship between poplar trees and fungi, assisting plants in coping with adverse environmental conditions through different mechanisms [23]. In this study, we explored the potential of AMF to enhance the salt tolerance of ‘84K’ poplar trees and elucidated the potential underlying physiological mechanisms. The results of this study offer valuable insights into the AMF−mediated physiological changes in different tissues of poplar trees under salt stress conditions.

### 3.1. Effects of Salt Stress and AMF Inoculation on Plant Growth and Quality

AMF colonization can occur in poplar trees under all salt concentrations. Generally, plants with thicker roots, fewer branches, and fewer root hairs have a higher dependence on mycorrhizal symbiosis. Severe salt stress can reduce the mycorrhizal colonization rate, which is consistent with previous studies [24]. Additionally, it has been noted that the benefits of mycorrhizal fungal symbiosis to plants are more pronounced under moderate than severe salt stress, possibly due to the inhibitory effects of NaCl [25]. Glomus−associated soil protein is a kind of glycoprotein secreted into soil by AMF. It can serve as a biological indicator for assessing forest soil quality and evaluating the stability of AMF in binding soil particles and promoting soil aggregation [26]. Our investigation found that whether under salt stress or not, the introduction of exogenous AMF into the rhizosphere soil of poplar effectively increased the GRSP content. AMF can effectively increase the GRSP level of *Medicago sativa* [27] and sorghum [28] under salt stress, greatly improving the stability of soil aggregates. Therefore, AMF can improve soil aeration and water retention by secreting GRSP, sequester Na^+^, and prevent salt transport in plants [29].

Plants inoculated with AMF alleviated the growth inhibition caused by salt stress at various salinity levels. This beneficial effect of AMF symbiosis on plant growth has also been observed in plants such as rosemary [30] and *Stevia rebaudiana Bertoni* [24], where AMF symbiosis alleviated the reduction in plant weight and root−to−shoot ratio caused by salt stress. The improvement in plant growth under salt stress conditions, attributed to mycorrhizal symbiosis, may be due to the enhanced water uptake and improved nutrient acquisition promoted by the mycorrhizal outer root mycelium [31]. Salt stress can reduce the relative water content of poplar leaves [32]. The mycelium aids the host plant in absorbing water from the soil and transferring it to the roots, improving RWC. The presence of mycorrhizae can also regulate the switch between the apoplastic and symplastic water transport pathways, enabling plants to better respond to stem water requirements [33]. A previous study found that AMF can induce lignin deposition [34]. Our study also observed an increase in the lignification of poplar stems after AMF inoculation, which corresponded to an increase in peroxidase activity. Enhanced cell wall synthesis in the root symbiotic zone under salt stress is an important defense response of plants, leading to alterations in plant cell wall components [35]. Previous research also reported that AMF inoculation significantly increased the cellulose content in vetiver roots, which is consistent with our findings [36]. Cellulose and lignin synergistically improve the structure and function of plant cell walls, and AMF enhances the strength of poplar cell walls, promoting plant growth and development.

### 3.2. Effects of AMF Inoculation on Photosynthesis under Salt Stress

Salt stress can disrupt the chloroplast structure of poplar, resulting in a decrease in or loss of their ability to absorb CO_2_ [37]. Previous studies have demonstrated that AMF can effectively enhance the salt tolerance of plants such as *Ricinus communis* [38] and *Gleditsia sinensis* [14] by altering their photosynthetic characteristics. Chlorophyll fluorescence parameters serve as indicators of plant response under stressful conditions [39]. In the PLS−PM model constructed in this study, it was found that mycorrhiza can alleviate the negative effects of stress on plant growth by increasing transpiration flux, thereby enhancing the gas exchange capacity and actual light energy conversion efficiency. AMF can alleviate the decrease in stomatal conductance caused by salt stress and promote leaf gas exchange, possibly due to the increase in nutrient and water absorption promoted by AMF [40]. F_v_/F_m_, considered a reliable diagnostic indicator of damage from environmental stress [41], was significantly reduced under salt stress in our study, indicating damage to the photochemical efficiency of PSII in the light (F_v_/F_m_) and the effective quantum yield of photosystem II (Y(II)). However, AMF effectively mitigated this reduction. Salt stress can damage the photosystem II reaction centers and disrupt electron transport in photosynthetic structures. AMF can improve photosynthesis and PS II functionality under stress conditions by enhancing water absorption, stabilizing the chloroplast and membrane structure, increasing the RuBisCO content, and reducing the accumulation of free radicals [42].

### 3.3. Effects of AMF Inoculation on Antioxidant Enzymes under Salt Stress

Salt stress can induce the production of reactive oxygen species (ROS), resulting in oxidative stress. ROS can oxidize cellular components, causing damage to lipids, proteins, and nucleic acids, as well as cell membrane destruction [43,44]. Malondialdehyde is the end product of lipid peroxidation, and its concentration reflects the extent of damage. In our study, the MDA content in plant tissues increased with salinity, confirming the enhancement of lipid peroxidation under salt stress. The antioxidant system is believed to play a crucial role in mediating tolerance to environmental stress. Plants employ defense systems by altering enzyme activity to scavenge reactive oxygen species. Superoxide dismutase catalyzes the dismutation of superoxide radicals into H_2_O and oxygen [45]. Peroxidase pairs reduce free radical concentrations and prevent lipid peroxidation [46]. Catalase also participates in the elimination of H_2_O_2_ after reacting with superoxide radicals mediated by SOD [47]. AMF inoculation can enhance the activity of antioxidant enzymes in plants [48]. Our study also confirmed that, under salt stress, AMF enhanced the activity of antioxidant enzymes (POD, CAT, and SOD) to counteract oxidative stress. Mycorrhizal symbiosis induces changes in plant signal transduction and enzyme synthesis genes [49]. AMF reduces stress−induced oxidative damage by enhancing antioxidant enzyme activity.

### 3.4. Effects of AMF Inoculation on Elemental Uptake under Salt Stress

Studies on the nutrient dynamics between AMF and host plants revealed that host plants supply carbon sources to AMF, stimulating the absorption and transfer of nitrogen and phosphorus elements from AMF to the host [50]. Mycorrhizae can enhance the transport rate of phosphorus through cytoplasmic recycling to regulate the salt tolerance mechanism of plants [51]. Mycorrhizal plants often exhibit higher nitrogen fixation capacity. Differential protein expressions in AMF under salt stress were significantly associated with nitrogen metabolism [52]. In our study, primary root inoculation with AMF under high salinity resulted in increased plant nitrogen and phosphorus concentrations. The main mechanism of AMF in enhancing host salt tolerance includes the mycelium and fine roots improving nutrient absorption and transportation [53].

Excessive soil Na^+^ concentration inhibits the absorption of K^+^, Ca^2+^, and Mg^2+^ by plants, resulting in reduced nutrient uptake and accumulation in plant tissues, ultimately limiting plant growth [54].

The long−distance transport of Na^+^ from roots to shoots occurs through the xylem, and the aboveground parts of plants are the main sites where toxicity occurs due to Na^+^ accumulation [55]. AMF inoculation has been demonstrated to reduce both Na^+^ concentration and the rate of transport to the shoots [56]. Potassium plays various roles in plant metabolism, including stomatal movement, protein synthesis, and enzyme activation. Studies have demonstrated that AMF can increase the K^+^ concentration in plants like *Ocimum basilicum* and *Satureja hortensis* [57]. Increasing the K^+^ concentration while accelerating Na^+^ removal strategies can assist plants in withstanding salt stress [58]. It is important to note that Na^+^ competes with K^+^ for binding sites essential to various cellular activities. Reducing Na^+^ concentration and maintaining Na^+^ and K^+^ balance in the cytoplasm are crucial factors for promoting plant growth under salt stress. In our study, under severe salt stress, AMF inoculation significantly reduced Na^+^ concentration, increased K^+^ concentration, and improved poplar’s salt stress tolerance.

Cucumber plants inoculated with AMF under stress will increase Ca^2+^ absorption [59]. Consistent with previous findings, the inoculation of AMF in salt−stressed poplar resulted in a significant increase in the Ca^2+^ concentration within the plant’s tissues. Ca^2+^ serves as an important cellular messenger in plant growth, enhancing enzyme activity and actively participating in cellular energy metabolism [60]. AMF contributes to the elevation of Ca^2+^ concentration, stabilizing the membrane and cell wall structure, preventing the influx of Na^+^, enhancing hydraulic conductivity, and activating the Ca^2+^ signaling pathway. These actions collectively serve to safeguard the photosynthetic tissue and protect plant physiological cell functions from salt stress [61]. Mg^2+^ also plays a vital role in the metabolism of laser synthesis and the subsequent transport of these products to roots. However, in our study, no significant differences in Mg^2+^ concentration were observed in mycorrhizal plants. This outcome may be attributed to variations in Mg^2+^ absorption capacity among different AMF species and the limited ability of mycorrhizal plants to absorb magnesium.

### 3.5. Influence of AMF Inoculation on Osmotic Regulation under Salt Stress

Relative electrolyte leakage is an important indicator for assessing membrane damage in plants under various stresses, and low REL values are indicative of less severe membrane damage [62]. In our experiment, AMF inoculation maintained lower levels of REL in leaves, promoting better plant growth under stress conditions. The disruption of plant cell membranes under stress conditions profoundly affects the osmotic system. Proline, serving as a regulatory substance, plays a crucial role in stabilizing the molecular structure of plants when subjected to high−salt environments [63]. Our study observed a significant increase in the proline content within plants under salt stress, followed by a decrease after AMF inoculation. This observation suggests that AMF can effectively alleviate salt stress in poplar plants. This finding aligns with a previous study that reported that AMF led to a reduction in proline content and stress mitigation in *Gleditsia sinensis* [14]. Glycine betaine functions as an osmoprotectant and operates by inhibiting the production of reactive oxygen species [64]. In our study, and AMF inoculation significantly increased the GB content in plants under salt stress. This increase corresponded with enhanced enzyme levels within the antioxidant system. It represents one of the mechanisms by which the quaternary ammonium compound GB counteracts oxidative stress in plants. AMF colonization can effectively alleviate damage under salt stress and promote the growth of seedlings by affecting proline and glycine betaine on roots.

Soluble sugars can serve as a source of nitrogen or carbon during stress recovery, and they also play an important role in osmoprotection by maintaining cell osmotic balance and free radical scavenging [65]. In our study, we observed considerable soluble sugar levels under severe salt stress, which enhanced the accumulation of nutrients and synthesis of essential compounds in the host. In mycorrhizal plants, the root sucrose content is higher, contributing to trehalose and lipid synthesis, as well as the production of the large extramatrical mycelium, and supporting the respiratory demands of the root systems [66]. Previous research demonstrated that AMF inoculation enhanced sucrose metabolism and regulation in maize under low−temperature stress [67]. In our study, the inoculation of mycorrhiza under salt stress promoted the accumulation of sucrose in plants. Additionally, our PLS−PM model indicated that sucrose and glycine betaine were significantly affected by mycorrhiza, contributing to the growth enhancement of aboveground plants parts. Under salt stress conditions, AMF can maintain osmotic balance in plants by regulating proline levels, enhancing the production of osmoprotectants, improving antioxidant capacity, and facilitating element uptake.

## 4. Materials and Methods

### 4.1. AMF Inoculum

*Rhizophagus irregularis* (BGC JX04B), provided by Changjiang University, was used as an inoculum. AMF inoculum consisted of AMF spores (each pot inoculated with 50 g of inoculum contained 600 spores), mycelium, root−broken soil, and soil.

### 4.2. Experimental Design

The experiment was conducted from June 2022 to October 2022 at the Sunlight Greenhouse of the Sanqingyuan Nursery of Beijing Forestry University (Figure 8). We chose healthy and similar 84K tissue culture seedlings with a plant height of 7.74 cm and a four−leaf stage. The topsoil (0–20 cm) and peat were mixed in a 2:1 ratio, sieved through a 20−mesh sieve, and added to pots with an outer diameter of 26.5 cm at a weight of 4.5 kg per pot. At the time of transplantation, AMF microbial inoculum, at a rate of 50 g per pot, was added at 2/3 of the height of the culture medium to ensure contact with the root system. The experiment consisted of two factors: (1) inoculation (AM) and non−inoculation (NM) treatments and (2) varying levels of salt stress (S0: 0 mM, S1: 100 mM, and S2: 200 mM). For each treatment, we had 12 replicates, resulting in a total of 2 × 3 × 12 = 72 pots. Watering was managed according to regular cultivation practices without exogenous nutrient solution treatment. After 90 days of transplantation, salt stress treatments were implemented through the daily addition of 50 mL of NaCl solution for 10 consecutive days at three concentration levels. After completing the stress treatments, the plants were further managed for 30 days before harvest.

### 4.3. AMF Root Colonization and Soil Spore Density

After harvest, the roots were delicately washed and preserved in FAA fixative solution following the method established by Phillips [68]. Subsequently, the samples were immersed in a 10% KOH solution and heated at 90 °C in a water bath for 30–60 min. Following softening, the samples were soaked in 2% HCl for 5–10 min and then stained with 0.05% Trypan Blue in lactoglycerol (40 mL lactic acid, 50 mL glycerol, and 10 mL distilled water). Thirty root segments, each measuring 1 cm, were selected from each sample and placed on a microscope slide for mounting with lactoglycerol. Photographs were taken under an optical microscope, and the colonization intensity was measured using the magnified intersections method [69]. The collected soil samples were mixed evenly, and 10 g subsamples were weighed to determine the spore content. AMF spores were centrifuged through modified wet sieving and the sucrose centrifugation method [70]. After centrifugation, the 60% sucrose solution was poured directly onto filter paper for filtration; fresh viable spores were counted by observing the filter paper under a microscope, and the number of AMF spores per unit weight of soil was determined.

AMF spore density (number/g) = number of spores in 10 g soil/10 g soil

### 4.4. Plant Growth Quality

Leaf sampling was conducted by harvesting the fully expanded fourth or fifth leaf from the top of each plant. The relative water content of the leaves was measured. RWC (%) was calculated as (FW − DW)/(TW − DW) × 100, where FW represents the fresh weight of the leaf, TW represents the weight of the sample after soaking in distilled water until fully saturated, and DW is the weight of the sample after drying in a 105 °C oven to a constant weight. For biomass determination and calculation of the root−to−shoot ratio, six samples were selected from each group and divided into roots, stems, and leaves. The samples were dried at 105 °C until a constant weight was achieved. Stem samples underwent enzyme−linked immunosorbent assay (ELISA) using antibodies specific to lignin and cellulose molecules for detection [71].

### 4.5. Detection of Glomalin−Related Soil Protein

Glomalin−related soil protein was extracted using a modified version of the method proposed by Wright et al. [72]. The method for the extraction of easy−to−extract glomalin−related soil protein was as follows: 0.5 g of air−dried soil was weighed and placed in a centrifuge tube and 4 mL of 20 mmol/L sodium citrate extract (pH = 7.0) was added. The sample was extracted at 103 kPa and 121 °C for 30 min, depressurized, and centrifuged at 4000 rpm for 6 min; the supernatant was collected for analysis. The method for the extraction of total glomalin−related soil protein was as follows: 0.1 g of air−dried soil was weighed and placed in a centrifuge tube and 4 mL of 50 mmol/L sodium citrate leaching solution (pH = 8.0) was added. The sample was extracted at 103 kPa and 121 °C for 60 min, depressurized, and centrifuged at 4 °C and 4000 rpm for 6 min. The supernatant was collected, fresh leaching solution was added, and the extraction was repeated at least twice until the extracted supernatant no longer showed the typical GRSP yellow–brown color. Finally, the total collected supernatant was combined and thoroughly mixed for analysis.

The protein content was determined using the Coomassie brilliant blue method: 0.5 mL of the solution to be measured was collected and mixed with 5 mL of Coomassie brilliant blue G−250 dye. The absorbance was recorded at 595 nm by a microplate reader. Using bovine serum albumin as the standard substance, a standard curve was established using the same method. Finally, the protein concentration and GRSP content in the solution were calculated from the standard curve, and the results are expressed as mg of protein in 1 g of soil [73].

### 4.6. Photosynthetic Physiology

From each group, six randomly selected plants were used for leaf measurements, specifically the fully expanded fourth or fifth leaf, taken within one week before harvest. A portable chlorophyll fluorometer (Junior PAM) was employed to measure chlorophyll fluorescence parameters of the leaves following a 30 min dark adaptation period. Each plant was measured 5–8 times, during which we recorded both the minimum fluorescence (F_o_) and maximum fluorescence (F_m_) of the leaves. The maximum quantum yield of PSII (F_v_/F_m_) was calculated using F_v_ = F_m_ − F_o_, while the effective quantum yield of PSII (Φ_PSII_) was calculated using Φ_PSII_ = (F_m_’ − F)/F_m_’ [74]. Under light−adapted conditions, the non−photochemical quenching was determined using the formula qN = 1 − (F_m_’ − F_o_’)/(F_m_ − F_o_’) [75], and the photochemical quenching was calculated using the formula qP = (F_m_’ − F)/(F_m_’ − F_o_’) [74]. The leaf photosynthetic parameters, including net photosynthesis rate, transpiration rate, and intercellular CO_2_ concentration, were measured using a photosynthesis analyzer (Li−6400XT). Each plant was measured 5–8 times, and the average values were recorded.

### 4.7. Antioxidant Defense

The lipid peroxidation level was determined by measuring malondialdehyde, which can condense with thiobarbituric acid (TBA) to generate a red product with strong absorption at 532 nm. The capacity of superoxide dismutase to reduce nitroblue tetrazolium (NBT) was measured at 560 nm. The ability of catalase to decompose hydrogen peroxide (H_2_O_2_) was assessed by measuring the change in absorbance at 240 nm. The activity of peroxidase, which catalyzes the oxidation of substrates by peroxides, was assessed by measuring the oxidation rate of the substrate at 470 nm. The respective assay kits (Solarbio Science & Technology Co., Ltd., Beijing, China) were employed to measure the aforementioned indicators according to the manufacturer’s instructions. For the calculation of the physiological indexes of AM inoculation, the formula was (Δ Index_(S0)_ + Δ Index_(S1)_ + Δ Index_(S2)_)/3.

### 4.8. Nutrient Content

After drying the roots, stems, and leaves of poplar trees at 105 °C to a constant weight, they were ground into a uniform powder and sieved through a 20 μm mesh. The nitrogen content was determined using the Kjeldahl method [76], the phosphorus content was measured using the vanadium–molybdenum yellow colorimetric method, and the potassium content was analyzed using flame atomic absorption spectrometry. The dried powder was dissolved in a digestion solution containing nitric acid and hydrogen peroxide, and the sample was heated for digestion. After diluting the sample, the concentrations of elements were determined using inductively coupled plasma mass spectrometry (ICP−MS) [68].

### 4.9. Osmotic Equilibrium

Leaves were sampled from the fourth or fifth fully expanded leaf of each pot. The leaves were gently washed with tap water to remove surface dust and then rinsed with deionized water. Circular leaf discs with a diameter of 0.6 cm were prepared. The leaf discs were placed in deionized water and subjected to vacuum infiltration. Subsequently, they were oscillated at 25 °C and 200 rpm for 3 h. The electrical conductivity (EC_1_) of the solution was measured using a conductivity meter. The leaf discs were subsequently immersed in a boiling water bath for 10 min to denature the tissues. After cooling, the electrical conductivity (EC_2_) was measured. The relative electrolyte leakage was calculated using the formula REL (%) = 100 × EC_1_/EC_2_. Glycine betaine was detected using an enzyme−linked immunosorbent assay (ELISA) method. After the substrate reacted with the samples, the concentration of the sample was determined at a wavelength of 450 nm using an ELISA. Proline content was determined by measuring the absorbance of the purple product formed by the reaction between proline and ninhydrin at 520 nm. The anthrone colorimetric method was utilized to quantify soluble sugars in plants. After the hydrolysis of sucrose, the reaction with anthrone was conducted, and the color was measured at 480 nm. The detection of proline, soluble sugars, and sucrose was carried out using the corresponding assay kits (Solarbio Science & Technology Co., Ltd. Beijing, China).

### 4.10. Statistical Analysis

The experimental data were subjected to analysis of variance (ANOVA) using R (v4.2.3), followed by Duncan’s multiple−range test for mean comparison. The ANOVA was performed to determine the significance of salt stress, inoculation treatment, and the interaction between salt stress and inoculation. Principal component analysis was conducted using the vegan package to show the effects of salt and mycorrhizal inoculation on poplar. Partial least squares path modeling was performed using the plspm package to model the relationships among mycorrhiza, the salt tolerance of poplar, and the physiological indexes of poplar. All the graphics were created using the ggplot2 package. Significance for all the statistical tests was set at *p* < 0.05.

## 5. Conclusions

AMF (*Rhizophagus irregularis*) effectively mitigated the deleterious effects of salt stress on the growth of poplar seedlings through multiple mechanisms. AMF significantly stimulated plant growth and optimized plant development by augmenting photosynthesis and water use efficiency. AMF inoculation increased the uptake of carbon, nitrogen, and phosphorus, diminished the absorption of Na^+^, and upheld ion equilibrium within the plant. Additionally, AMF enhanced the activity of antioxidant enzymes (POD, CAT, and SOD) in scavenging reactive oxygen species and reducing oxidative damage to cell membranes. These results indicate a well−established symbiotic relationship between AMF and poplar seedling roots, resulting in reduced levels of proline and MDA in mycorrhizal plants, indicative of improved salt tolerance. These findings highlight the significant potential of AMF for afforestation and forest restoration in salt−affected areas, including coastal regions.

## Figures and Tables

**Figure 1 plants-13-00233-f001:**
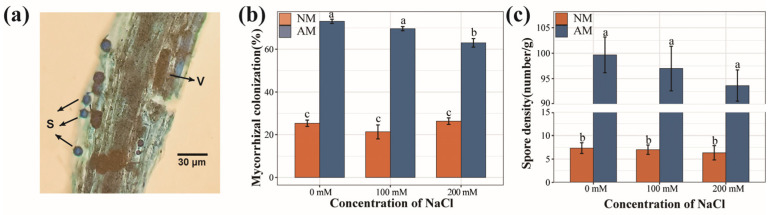
AMF colonization of poplars. (**a**) Mycorrhiza microstructure (arrows: S, spores; V, vesicles). (**b**) Mycorrhizal colonization rate of poplar under different levels of salt stress. (**c**) Spore density in rhizosphere soil of poplars under different levels of salt stress. NM, non-inoculated treatment; AM, AM-inoculated treatment. Different letters on top of the bars indicate significant differences (*p* < 0.05). All data are presented as means ± standard errors.

**Figure 2 plants-13-00233-f002:**
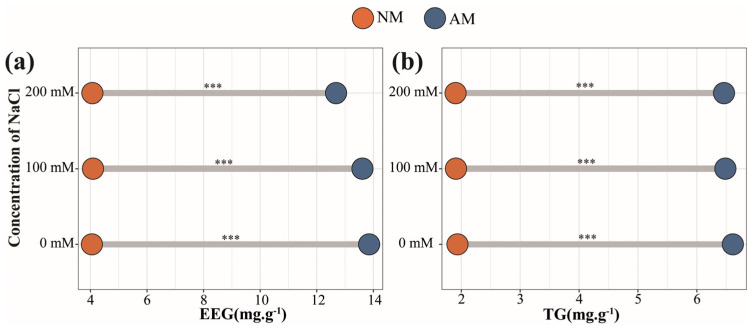
Glomalin−related soil protein (GRSP) contents in poplar rhizosphere soil. (**a**) EEG contents under salt stress. (**b**) TG contents under salt stress. NM, non−inoculated treatment; AM, AM−inoculated treatment. ***, *p* < 0.001. All data are presented as means ± standard errors.

**Figure 3 plants-13-00233-f003:**
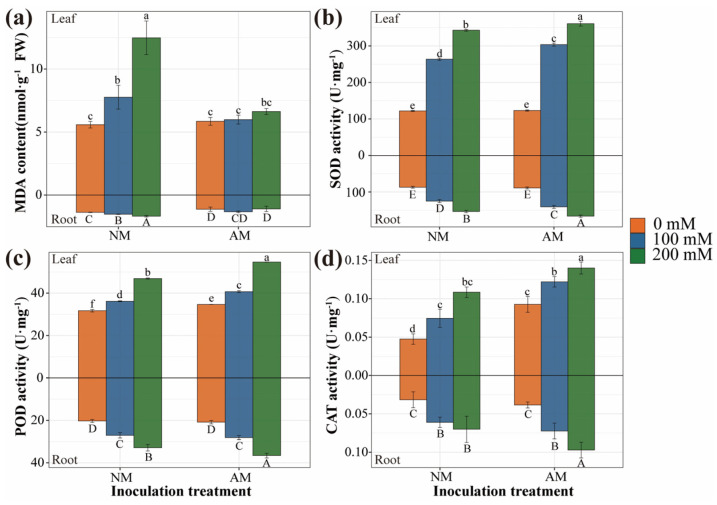
Effect of AMF inoculation on antioxidation under salt stress. (**a**) MDA content of poplar. (**b**) SOD activity of poplar. (**c**) POD enzyme activity of poplar. (**d**) CAT enzyme activity of poplar. NM, non−inoculated treatment; AM, AM−inoculated treatment. The lower case at the top (Leaf) and the capital letters at the bottom (Root) of the bar chart indicate significant differences (*p* < 0.05). All data are presented as means ± standard errors.

**Figure 4 plants-13-00233-f004:**
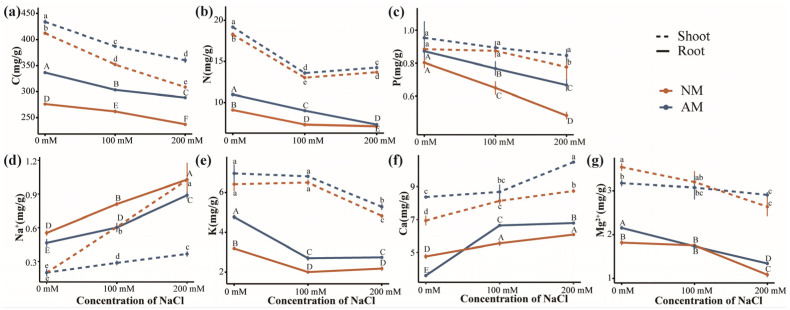
Effect of AMF inoculation on nutrient absorption of poplar under salt stress. (**a**–**c**) C, N, and P contents of poplar. (**d**–**g**) The contents of Na^+^, K^+^, Ca^2+^, and Mg^2+^ in poplar. NM, non−inoculated treatment; AM, AM−inoculated treatment. Different letters on top of the bars indicate significant differences (*p* < 0.05). All data are presented as means ± standard errors.

**Figure 5 plants-13-00233-f005:**
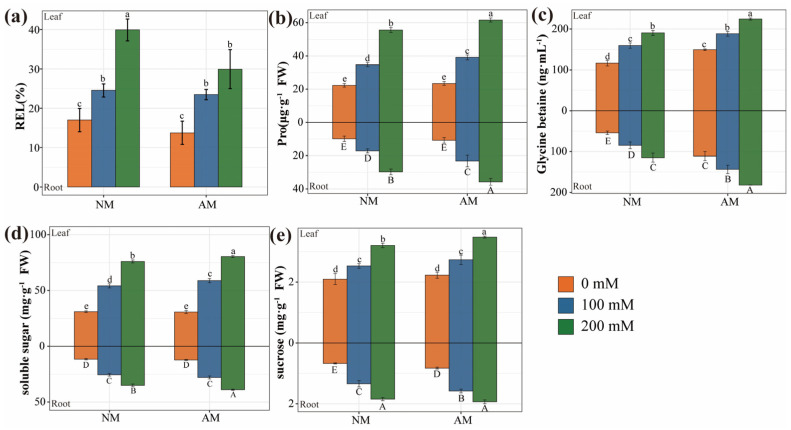
Effect of AMF inoculation on osmotic balance of poplar under salt stress. (**a**) Leaf REL of poplar. (**b**) Pro content in poplar. (**c**) GB concentration of poplar. (**d**) Soluble sugar content of poplar. (**e**) Sucrose content of poplar. NM, non−inoculated treatment; AM, AM inoculated treatment. The lower case at the top (Leaf) and the capital letters at the bottom (Root) of the bar chart indicate significant differences (*p* < 0.05). All data are presented as means ± standard errors.

**Figure 6 plants-13-00233-f006:**
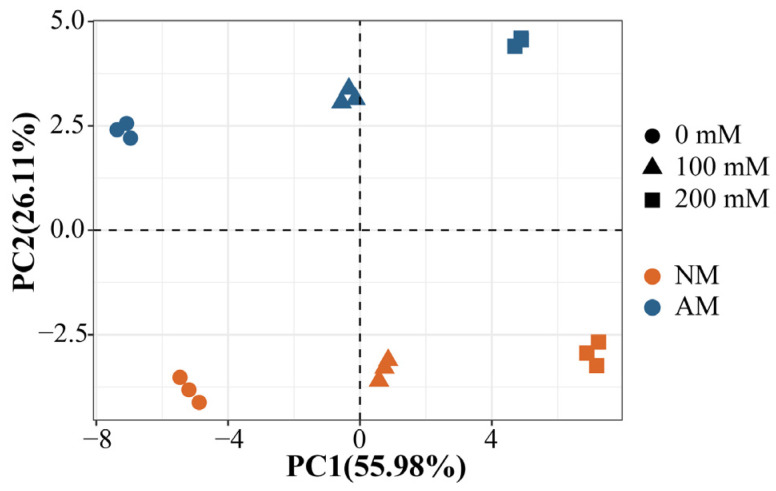
PCA diagram of two principal components of poplar. PC1, principal component 1; PC2, principal component 2. NM, non−inoculated treatment; AM, AM−inoculated treatment.

**Figure 7 plants-13-00233-f007:**
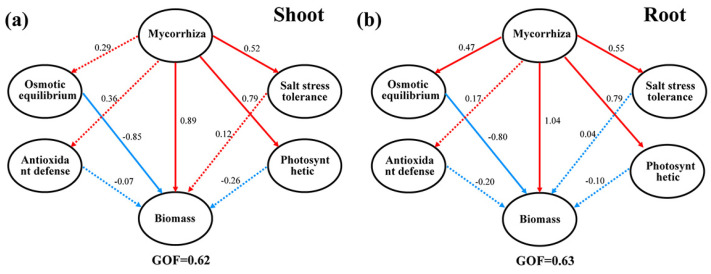
Correlations among mycorrhiza, salt tolerance, oxidation resistance, photosynthesis, infiltration, and biomass according to partial least squares path modeling. (**a**) Shoot. (**b**) Root. Blue and red arrows indicate positive and negative path coefficients, respectively, and solid and dashed lines indicate significant and nonsignificant path coefficients, respectively (*p* < 0.05).

**Figure 8 plants-13-00233-f008:**
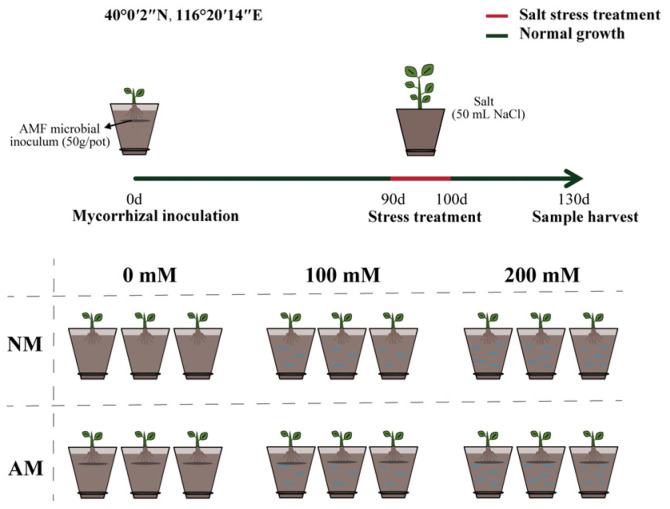
Experimental design of AMF−inoculated poplar under salt stress.

## Data Availability

All data in this study can be found in the manuscript.

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
