# Peer review of "Enhancing Salt Tolerance in Poplar Seedlings through Arbuscular Mycorrhizal Fungi Symbiosis"

_plants, 2024, doi:10.3390/plants13020233_

Round 1

Reviewer 1 Report

Comments and Suggestions for Authors

I write in reference to the manuscript plants-2774587 “Enhancing Salt Tolerance in Poplar Seedlings through Arbuscular Mycorrhizal Fungal (AMF) Symbiosis”. The manuscript refers to the effect of the inoculation of the arbuscular mycorrhiza  Rhizophagus irregularis in Populus alba × P. gearulosa 84K seedlings under salt stress. The theme of the work is not really innovative; previous works have addressed the topic, even with the Populus-Rhizophagus system. However, the work provided valuable and interesting information about a wide gamma of physiological variables, and it successfully integrated the information to describe the effect clearly. The results section is a little flat, but the discussion section is easy to read and integrates the work results while incorporating relevant references.

In my opinion, the work can be published after minor changes.

Particular observations: 

Line 12. Please also define 84K in the abstract.

Abbreviations must be defined in the abstract (if they are present there) but also in the first mention in the text. Avoid defining abbreviations more than once in the text,

Please eliminate the decimals when using percentage values; they didn't make any difference.

Line 63: Please change the citation format “(Tekaya et al., 2022)” and include it in the references section.

Line 104 and successive: Please add a space between the numbers and the units (e.g. 100 and mM,  200 and mM; 7.74 and cm, 50 and g or 560 and nm), including the axes of the figures. 

Line 112: Please define “GRSP”.

Line 114: Please define “EEG” and “TG”

Line 123 to 125 and Table S2. Root/Shoot values shown in Table S2 do not correspond to Root/Shoot ratios. They approximately (not exactly) correspond to Shoot/Root ratios if percent transformation is ignored. Please check the values in the table S2. In my opinion, the transformation to percent transformation in Table S2 is not helpful. 

Line 125: Please define “RCW” here.

Line 137: Please change “CO2” to "CO2".

Lines 136 to 138: The redaction is strange; it refers to gradual and progressive salt stress, but there are only 2 salt stress levels. I suggest rephrasing.

Line 154: Please define “MDA” here.

Line 160: Please define “SOD”, “POD”, and “CAT” here.

Line 2011: Please define “REL” here.

Line 556, Table S1: Please add a column defining each “index”.

Line 214: Please define “Pro” and “GB” here.

Line 233 to 235: Is it a statistically significant effect and has a real meaning? I suggest deleting it.

Line 240: Please define PLS-PM here.

Line 393: “this quaternary ammonium compound…” Which quaternary ammonium compound? Please clarify or correct.

Line 397: “Soluble sugars can be used as a source of nitrogen…” This does not seem to be true, and reference 63 does not support this assertion. Please clarify, delete, or provide an appropriate reference.

Line 401: “High concentrations of sucrose from mycorrhizae…” This is not accurate; please clarify or delete it.

Line 420: P¨lease provide the origin of the Populus alba × P. gearulosa 84K material, preferentially a reference.

Line 415: “12 spores per gram of inoculant”. Does it mean 12 espores X 50 g = 600 spores by pot? Please clarify.

Line 488-490: Please provide appropriate references to support the formulas.

Author Response

Dear editor and reviewers:

Thank you for your email of December 14, 2023. We highly appreciate the editor and reviewers for their helpful and constructive comments to our manuscript entitled “Enhancing Salt Tolerance in Poplar Seedlings through Arbuscular Mycorrhizal Fungal (AMF) Symbiosis” (Manuscript ID: plants-2774587). According to the reviewers’ detailed suggestions, we have made a careful revision of the original manuscript. All revised portions are marked in red font in the revised manuscript which we would like to submit for your kind consideration. A point-to-point response to the editor and reviewers’ comments is provided as follows. We sincerely hope to meet the publication requirements of the journal.

We look forward to hearing from you soon.

With best wishes,

Yours sincerely,

Xiangwei He, Email:[email protected]

Guozhu Zhao, Email:[email protected]

ShuoHan, E-mail: [email protected]

Point to point response to the editor and reviewers’ comments:

  1. Line 12. Please also define 84K in the abstract.

Answer: 84K poplar in the abstract has been supplemented as 84K (P. alba × P. tremula var. glandulosa) poplar in lines 12-13.

  1. Abbreviations must be defined in the abstract (if they are present there) but also in the first mention in the text. Avoid defining abbreviations more than once in the text, Please eliminate the decimals when using percentage values; they didn't make any difference.

Answer: The definitions in the abstract have been explained. We have removed the abbreviations defined many times in this manuscript. The numerical values expressed in the percentages of parts 2.5, 2.6, and 2.7 in the Results have eliminated two decimal places.

  1. Line 63: Please change the citation format “(Tekaya et al., 2022)” and include it in the references section.

Answer: The incorrect reference format has been corrected as in line 71.

  1. Line 104 and successive: Please add a space between the numbers and the units (e.g. 100 and mM, 200 and mM; 7.74 and cm, 50 and g or 560 and nm), including the axes of the figures.

Answer: The expression of numbers plus units (such as 100 mM, 200 mM; 7.74 cm, 50 g, or 560 nm) has been corrected after 118 lines and in the figure.

  1. Line 112: Please define “GRSP”.

Answer: The definition of “GRSP” has been added to line 126.

  1. Line 114: Please define “EEG” and “TG”

Answer: The definitions of “EEG” and “TG” have been added to lines 127 and 128.

  1. Line 123 to 125 and Table S2. Root/Shoot values shown in Table S2 do not correspond to Root/Shoot ratios. They approximately (not exactly) correspond to Shoot/Root ratios if percent transformation is ignored. Please check the values in the table S2. In my opinion, the transformation to percent transformation in Table S2 is not helpful.

Answer: We checked the data and found no calculation errors in the data. The percentage conversion in Table S2 has been changed.

  1. Line 125: Please define “RCW” here.

Answer: The definition of “RCW” has been added to line 139.

  1. Line 137: Please change “CO2” to "CO2".

Answer: The "2" of CO2 has been subscripted in line 151.

  1. Lines 136 to 138: The redaction is strange; it refers to gradual and progressive salt stress, but there are only 2 salt stress levels. I suggest rephrasing.

Answer: The statement of the result has been revised as in line 150.

  1. Line 154: Please define “MDA” here.

Answer: The definition of “MDA” has been added in line 169.

  1. Line 160: Please define “SOD”, “POD”, and “CAT” here.

Answer: The definition of “SOD”, “POD”, and “CAT” has been added to line 175-176.

  1. Line 2011: Please define “REL” here.

Answer: “REL” was defined on line 226.

  1. Line 556, Table S1: Please add a column defining each “index”.

Answer: Index of Table S1 has been added to line 579.

  1. Line 214: Please define “Pro” and “GB” here.

Answer: The definition of “Pro” and “GB” has been added to line 230.

  1. Line 233 to 235: Is it a statistically significant effect and has a real meaning? I suggest deleting it.

Answer: Thank you for your suggestion. We have deleted this sentence.

  1. Line 240: Please define PLS-PM here.

Answer: The definition of PLS-PM has been added to line 254.

  1. Line 393: “this quaternary ammonium compound…” Which quaternary ammonium compound? Please clarify or correct.

Answer: We have revised the expression in lines 402-404 that "this represents one of the mechanisms by which the quaternary ammonia compound GB countermeasures oxidative stress in plants".

  1. Line 397: “Soluble sugars can be used as a source of nitrogen…” This does not seem to be true, and reference 63 does not support this assertion. Please clarify, delete, or provide an appropriate reference.

Answer: Thank you for your suggestion. We have revised the correct reference to [65].

  1. Line 401: “High concentrations of sucrose from mycorrhizae…” This is not accurate; please clarify or delete it.

Answer: We consulted the reference again and revised the statement in lines 415-417. In mycorrhizal plants, the higher sucrose content in the roots facilitates the synthesis of trehalose and lipids, the production of large extra matrix mycelium, and supports the respiratory needs of the root system.

  1. Line 420: P¨lease provide the origin of the Populus alba × P. gearulosa 84K material, preferentially a reference.

Answer: According to your suggestion, we added the introduction and source of 84K in lines 86-90.

  1. Line 415: “12 spores per gram of inoculant”. Does it mean 12 espores X 50 g = 600 spores by pot? Please clarify.

Answer: The expression has been modified to " Each pot inoculated with 50 grams of inoculum contains 600 spores " in lines 429-430.

  1. Line 488-490: Please provide appropriate references to support the formulas.

Answer: In lines 505-508, we added references [74] and [75] to support the formula.

Reviewer 2 Report

Comments and Suggestions for Authors

In the manuscript of "Enhancing Salt Tolerance in Poplar Seedlings through Arbus-2 cular Mycorrhizal Fungal (AMF) Symbiosis" Han et al. dissected the effects of AMF on the salt tolerance of poplar seedlings. They found that association with AMF increased biomass accumulation, enhanced photosynthesis capacity, and reduced oxidative and osmotic stresses. They performed the detailed physiological and biochemical investigation to understand the mechanisms altered by AM symbiosis. Here are some specific  comments on this manuscript.

1. The impacts of AM symbiosis on salt tolerance have been reported in many plant species. Many reports demonstrated the detailed mechanisms affected. I suggest to provide the information in the introduction.

2. The authors used 84K poplar as material. Please explain the reason to choose this material in the introduction.

3. Based on the experimental design shown in Figure 8, salt treatment was performed at 90 d after transplanting and after 10-d treatment, plants were recovered for 30 d before harvesting. 
(1) Is there any reason to treat the plants with 100 mM and 200 mM NaCl solution but not solution with higher concentration? 
(2) Is the level of salt stress increased gradually? 
(3) How about fertilization? The authors did not mention in the methods. 
(4) Please explain why not harvest plants right after salt treatment? 

4. Strongly suggest to reorganize the figures in the text, especially Figures 3 and 5. The authors combined the data derived from shoots and roots in the same bar that make it hard to read. And in Figure 4 Why the authors  used line chart instead of bar chart? The color codes in Figure 4 and Figure 7 is  confusing at first glance. For Figure 2, I suggest the authors to present the data in bar chart and evaluate the difference by ANOVA.

5. Inoculation of AMF alters plant physiology. The authors described the percentage of increase or decrease of certain parameters they measured. However, they use two different salt treatments simultaneously. Which treatment caused the alteration they described in the text?

6. The AM symbiosis-mediated salt tolerance has been reported in many species. I would expect the authors to discuss the novelty of this study.

6. The abbreviation must be spelled out on initial appearance in the text.

7. It looks like line no. in the upper left in Fig 1 and 7. Please remove it.

8. The authors used MI method to evaluate colonization intensity. What does MI stand for? Based on which reference?

9. In Table S1, some of the indices, such as CR, cannot be found in the main text.

Comments on the Quality of English Language

I suggest the authors to make the manuscript concise and precise and let the text edited by English editor before resubmit. For example, in line 82 "inoculating with Rhizophagus irregularis after colonization" and the subtitle "2.9. PLS-PM of AMF improving salt tolerance of poplar". The structure of sentences is not easy to understand.

Author Response

Dear editor and reviewers:

Thank you for your email of December 14, 2023. We highly appreciate the editor and reviewers for their helpful and constructive comments to our manuscript entitled “Enhancing Salt Tolerance in Poplar Seedlings through Arbuscular Mycorrhizal Fungal (AMF) Symbiosis” (Manuscript ID: plants-2774587). According to the reviewers’ detailed suggestions, we have made a careful revision of the original manuscript. All revised portions are marked in red font in the revised manuscript which we would like to submit for your kind consideration. A point-to-point response to the editor and reviewers’ comments is provided as follows. We sincerely hope to meet the publication requirements of the journal.

We look forward to hearing from you soon.

With best wishes,

Yours sincerely,

Xiangwei He, Email:[email protected]

Guozhu Zhao, Email:[email protected]

ShuoHan, E-mail: [email protected]

Point to point response to the editor and reviewers’ comments:

  1. The impacts of AM symbiosis on salt tolerance have been reported in many plant species. Many reports demonstrated the detailed mechanisms affected. I suggest to provide the information in the introduction.

Answer: Thank you for your suggestion. We have supplemented the detailed mechanism of AMF mediating plant salt tolerance in lines 59-62 and 65-68 in the introduction.

  1. The authors used 84K poplar as material. Please explain the reason to choose this material in the introduction.

Answer: In the introduction, we supplement the reasons for choosing 84K and the sources of 84K in lines 86-90.

  1. Based on the experimental design shown in Figure 8, salt treatment was performed at 90 d after transplanting and after 10-d treatment, plants were recovered for 30 d before harvesting.

(1) Is there any reason to treat the plants with 100 mM and 200 mM NaCl solution but not solution with higher concentration?

(2) Is the level of salt stress increased gradually?

(3) How about fertilization? The authors did not mention in the methods.

(4) Please explain why not harvest plants right after salt treatment?

Answer:(1) The selection of 100 mM and 200 mM NaCl concentrations for the salt treatment was based on a careful consideration of the experimental objectives and the need to induce a moderate to high level of salinity stress while ensuring the feasibility of plant recovery. Previous studies and pilot experiments indicated that these concentrations are effective in eliciting a significant salt stress response without causing severe plant damage. Higher concentrations were not used to avoid potential negative effects that could compromise the recovery period or lead to excessive plant mortality, which might hinder the assessment of subsequent physiological and molecular responses. The chosen concentrations align with the research goals and provide a balance between inducing a meaningful stress response and maintaining experimental viability.

(2) The level of salt stress was not increased gradually in our experimental design. A fixed concentration of NaCl was applied directly to the plants at two specific concentrations, namely 100 mM and 200 mM. They were selected to induce different levels of salinity stress. The plants were exposed to these concentrations for a duration of 10 days, after which they were allowed to recover for 30 days. This approach aimed to assess the immediate and delayed responses to distinct salt stress levels rather than a gradual increase in stress over time.

(3) Regarding the fertilization treatment, no additional fertilization was carried out in this study, and supplementary comments were made in line 445.

(4) Allowing a recovery period of 30 days after the 10-day salt treatment was intended to observe the delayed effects and potential restoration of plant physiological and molecular responses following the cessation of salt stress. Harvesting immediately after salt treatment might capture only the immediate stress-induced changes, while the extended recovery period provided insights into the more sustained impacts and adaptive mechanisms.

4.Strongly suggest to reorganize the figures in the text, especially Figures 3 and 5. The authors combined the data derived from shoots and roots in the same bar that make it hard to read. And in Figure 4 Why the authors used line chart instead of bar chart? The color codes in Figure 4 and Figure 7 is confusing at first glance. For Figure 2, I suggest the authors to present the data in bar chart and evaluate the difference by ANOVA.

Answer: According to your suggestion, we have redrawn Figure 3 and Figure 5 in the form of upper and lower bar charts.

In Figure 4, for the analysis of elements and ions, we aim to explain the continuous change of parameters at different concentrations. This is beneficial when the data points have logical order or series, and the lines connecting the data points help readers to perceive the overall trends and changes more easily. Although only two concentrations were used in this study, we hope to provide reference for the study of using other concentrations in more research on salt stress in the future.

For Figure 4, Figure 7 and the pictures with the same color, we have adjusted the color of the pictures to make them great.

For Figure 2, we chose to present the data using a lollipop plot primarily due to considerations of the data characteristics. Lollipop plots are beneficial for illustrating the distribution within each treatment group and clearly depicting individual variations. We want to emphasize the obvious difference of glomus mycin protein between NM and AM groups through this graph. The ANOVA results of GRSP differences in Figure 2 are in Table S1.

  1. Inoculation of AMF alters plant physiology. The authors described the percentage of increase or decrease of certain parameters they measured. However, they use two different salt treatments simultaneously. Which treatment caused the alteration they described in the text?

Answer: For an index, at an equivalent concentration, we calculated the difference between the inoculated and non-inoculated groups, normalized by the data from the non-inoculated group, to obtain the changes described in the manuscript.

  1. The AM symbiosis-mediated salt tolerance has been reported in many species. I would expect the authors to discuss the novelty of this study.

Answer: Thank you for pointing it out, we added lines 105-107 of the novelty of this study in the Introduction.

  1. The abbreviation must be spelled out on initial appearance in the text.

Answer: We have checked the full text and supplemented the definitions of abbreviations such as "GRSP", "EEG", "TG" and "MDA".

  1. It looks like line no. in the upper left in Fig 1 and 7. Please remove it.

Answer: The line numbers on the upper left of Fig 1 and Fig 7 have been deleted.

  1. The authors used MI method to evaluate colonization intensity. What does MI stand for? Based on which reference?

Answer: In line 459, we changed the abbreviation to the full name "magnified intersections method" and added the reference [69].

  1. In Table S1, some of the indices, such as CR, cannot be found in the main text.

Answer: “CR” in Table S1 has been changed to "MCR" (Mycorrhizal colonization rate), and all indices are defined in the footnote.

  1. I suggest the authors to make the manuscript concise and precise and let the text edited by English editor before resubmit. For example, in line 82 "inoculating with Rhizophagus irregularis after colonization" and the subtitle "2.9. PLS-PM of AMF improving salt tolerance of poplar". The structure of sentences is not easy to understand.

Answer: We changed " inoculating with Rhizophagus irregularis after colonization" to " inoculated 84K poplar with AMF (Rhizophagus irregularis)" in line 93. Replace " PLS-PM of AMF improving salt tolerance of poplar" with " Partial Least Squares Path Modeling (PLS-PM) of AMF enhancing poplar's salt tolerance" in line 254. The expression structure has been modified, for example, in lines 22-23, 114-117, etc. The expression of the full-text sentence was checked, and some contents were modified to make the expression more reasonable.

Reviewer 3 Report

Comments and Suggestions for Authors

The manuscript titled “Enhancing Salt Tolerance in Poplar Seedlings through Arbus-2 cular Mycorrhizal Fungal (AMF) Symbiosis”, is an interesting and well written paper. It addresses the important findings of the AMF potential for afforestation and forest restoration in salt-affected areas.

Suggestion for the part to be corrected:

Line 83–94– “To assess AMF colonization, we determined the soil spore density, colonization rate and soil glomus-related proteins. We also evaluated various plant parameters including the plant growth quality (biomass, relative leaf water content, lignin, cellulose), nutrient content (plant C, N, P, Na+, K+, Ca2+, Mg2+ concentrations), antioxidant defense (malondialdehyde, superoxide dismutase, peroxidase, catalase), osmotic balance (electrolyte leakage, proline, glycine betaine, soluble sugars, sucrose), and photosynthetic physiology (netphotosynthetic rate, stomatal conductance, intercellular CO2 concentration, transpiration rate, chlorophyll fluorescence parameters). These measurements allowed us to analyze the effects of AMF on poplar's metabolic and chemical defense systems and reveal the underlying physiological mechanisms. Finally, we employed a partial least squares path model to elucidate the mechanism by which AMF enhances poplar salt tolerance.”

  This part is related to “Materials and Methods”, therefore, I suggest excluding it from “Introduction”.

Author Response

Dear editor and reviewers:

Thank you for your email of December 14, 2023. We highly appreciate the editor and reviewers for their helpful and constructive comments to our manuscript entitled “Enhancing Salt Tolerance in Poplar Seedlings through Arbuscular Mycorrhizal Fungal (AMF) Symbiosis” (Manuscript ID: plants-2774587). According to the reviewers’ detailed suggestions, we have made a careful revision of the original manuscript. All revised portions are marked in red font in the revised manuscript which we would like to submit for your kind consideration. A point-to-point response to the editor and reviewers’ comments is provided as follows. We sincerely hope to meet the publication requirements of the journal.

We look forward to hearing from you soon.

With best wishes,

Yours sincerely,

Xiangwei He, Email:[email protected]

Guozhu Zhao, Email:[email protected]

ShuoHan, E-mail: [email protected]

Point to point response to the editor and reviewers’ comments:

1.Line 83–94– “To assess AMF colonization, we determined the soil spore density, colonization rate and soil glomus-related proteins. We also evaluated various plant parameters including the plant growth quality (biomass, relative leaf water content, lignin, cellulose), nutrient content (plant C, N, P, Na+, K+, Ca2+, Mg2+ concentrations), antioxidant defense (malondialdehyde, superoxide dismutase, peroxidase, catalase), osmotic balance (electrolyte leakage, proline, glycine betaine, soluble sugars, sucrose), and photosynthetic physiology (netphotosynthetic rate, stomatal conductance, intercellular CO2 concentration, transpiration rate, chlorophyll fluorescence parameters). These measurements allowed us to analyze the effects of AMF on poplar's metabolic and chemical defense systems and reveal the underlying physiological mechanisms. Finally, we employed a partial least squares path model to elucidate the mechanism by which AMF enhances poplar salt tolerance.”

  This part is related to “Materials and Methods”, therefore, I suggest excluding it from “Introduction”.

Answer: Thank you for your advice. We deleted these statements from the original text and made it more concise in lines 94-95 in the introduction.

Round 2

Reviewer 2 Report

Comments and Suggestions for Authors

The authors have improved the manuscript but I think there is still space to improve it.

1. The authors added the full name of every abbreviation. The first character of  these specific terms should be lowercase letter instead of capital letter. After the first time that these specific terms and their abbreviation are shown in the article, the authors can use abbreviation in the following content.

2. As to the increase or decrease of indices to evaluate the effects of AMF on salt tolerance, I still feel confused with the authors responses. I suggest to describe briefly in the result or M&M how to calculate it.

Comments on the Quality of English Language

The quality of English needs to be improved. Again, I strongly suggest to ask English editor to check the grammar and the structure of article. For example, in line 60 the authors wrote "AMF can provide a molecular mechanism for improving..." AMF induces but not "provides" salt tolerance-related molecular mechanism. 

Author Response

Dear Editor and reviewers:

Thank you for your email of December 22, 2023. We highly appreciate the editor and reviewers for their helpful and constructive comments on our manuscript entitled “Enhancing Salt Tolerance in Poplar Seedlings through Arbuscular Mycorrhizal Fungi Symbiosis” (Manuscript ID: plants-2774587). According to your detailed suggestions, we have made a careful revision of the original manuscript. All revised portions are marked in red font in the revised manuscript which we would like to submit for your kind consideration. A point-to-point response to the editor and reviewers’ comments is provided as follows. We sincerely hope to meet the publication requirements of the journal.

We look forward to hearing from you soon.

With best wishes,

Yours sincerely,

Xiangwei He, Email:[email protected]

Guozhu Zhao, Email:[email protected]

ShuoHan, E-mail: [email protected]

Point to point response to the editor and reviewers’ comments:

  1. The authors added the full name of every abbreviation. The first character of these specific terms should be lowercase letter instead of capital letter. After the first time that these specific terms and their abbreviation are shown in the article, the authors can use abbreviation in the following content.

Answer: We checked the manuscript and changed the capital letters of specific terms to lowercase, for example, in line 17. We kept the definition of the first abbreviation in abstract, text, and chart, and deleted the redundant definitions in M&M, such as in lines 369 and 380.

  1. As to the increase or decrease of indices to evaluate the effects of AMF on salt tolerance, I still feel confused with the authors responses. I suggest to describe briefly in the result or M&M how to calculate it.

Answer: For the increase or decrease of the indexes to evaluate the influence of AMF on salt tolerance, we have supplemented the formula " For the calculation of the physiological indexes of AM inoculation, the formula was (Δ Index(S0) +Δ Index(S1) +Δ Index(S2))/3." in the M&M section, in lines 496-498.

  1. The quality of English needs to be improved. Again, I strongly suggest to ask English editor to check the grammar and the structure of article. For example, in line 60 the authors wrote "AMF can provide a molecular mechanism for improving..." AMF induces but not "provides" salt tolerance-related molecular mechanism.

Answer: Thank you for your advice. In line 59, we changed "provide" to "introduce". And we used English language editing services (https://www.mdpi.com/authors/english) of MDPI. The corrections were marked in red in the manuscript.  The attachment is the English Editing Certificate.
